# Potential endocrine disrupting properties of toys for babies and infants

**Christian Kirchnawy[1]ᵒ, Fiona Hager[1]ᵒ, Veronica Osorio Piniella[1], Mathias Jeschko[1], Michael Washüttl[1], Johannes Mertl[1], Aurelie Mathieu-Huart[2], Christophe Rousselle[2] ***

**1** OFI, Austrian Research Institute for Chemistry and Technology, Vienna, Austria, **2** Risk Assessment Department, French Agency for Food, Environmental and Occupational Health Safety (ANSES), Maisons-Alfort, France

ᵒ These authors contributed equally to this work.
* christophe.rousselle@anses.fr

**Data Availability Statement:** All relevant data are within the paper and its Supporting Information files.

**Funding:** "This study was partly financed by the ANSES, and partly financed by Österreichische

## Abstract

Plastic toys mouthed by children may be a source of exposure to endocrine active substances. The purpose of this study was to measure hormonal activity of substances leaching from toys and to identify potential endocrine disruptors causing that activity. For this purpose, migration experiments of toys were conducted in saliva simulants. The CALUX® assays were used to detect (anti-) estrogenic and (anti-) androgenic activity of 18 toys. Chemical trace analysis–namely, GC-MS and HPLC-MS- was used to identify which compounds may be responsible for endocrine activity in the sample migrates. Nine out of 18 tested toys showed significant estrogenic activity. For two samples, the detected estrogenic activity could be well explained by detecting the known endocrine active substance bisphenol A (BPA). For all identified substances, including BPA, a risk assessment for human health was performed by comparing the exposure dose, calculated based on the determined substance concentration, to toxicological reference values. Using worst-case scenarios, the exposure to BPA by mouthing of the two estrogen active, BPA-containing toys could be above the temporary TDI that EFSA has calculated. This demonstrates that some toys could significantly contribute to the total exposure to BPA of babies and infants. For seven out of nine estrogen active samples, the source of the estrogen activity could not be explained by analysis for 41 known or suspected endocrine active substances in plastic, indicating that the estrogen activities were caused by currently unknown endocrine active substances, or by endocrine active substances that would currently not be suspected in toys.

## Introduction

Children, in particular, those below the age of 36 months, are considered particularly vulnerable to chemical substance exposure, due to physiological differences (e.g. less metabolism capacities compared to adults) and different exposure patterns, such as hand-to-mouth behavior [1]. Furthermore, developing periods constitute windows of susceptibility to some

Forschungsförderungsgesellschaft mbH (FFG), in the context of the project "XENOFREE" (Project 845811). The FFG was not involved in study design, collection, analysis or interpretation of data. This funding source does not alter the authors' adherence to PLOS ONE policies on sharing data and materials. ANSES, the second funding source, is part of the author team and was strongly involved in study design and interpretation of data.

**Competing interests:** The authors have declared that no competing interests exist.

compounds showing endocrine activity. This may explain the increased incidence of certain diseases such as neurodevelopmental disorders or effects on the reproductive tract [2].

Children are exposed in their daily life to multiple chemical compounds, *via* food, dust, personal care products and other consumer products [3,4]. Several studies based on observations on the mouthing behavior of children from 0 to 36 months confirm that children put a great diversity of objects into their mouths, including toys [1]. Plastic toys account for the majority of toys purchased in France. However, only few studies have assessed the risks associated with chemical substances present in plastic toys and children's equipment intended for infants and children up to three years [5]. Plastic toys are often made with complex mixtures of one or more polymers combined with multiple additives such as plasticizers, flame retardants, antioxidants . . .. As some of these constituents are not covalently bound to the polymers, plastics have been shown to release chemicals such as phthalates or UV filters that are known endocrine disrupters [4].

Toys are regulated in the European Union by a Directive which stipulates that toy manufacturers shall carry out an analysis of the chemical, physical, mechanical, electrical, flammability, hygiene and radioactivity hazards that the toy may present and assess the potential exposure to these hazards. In terms of chemical safety, the directive prohibits the use of carcinogenic, mutagenic and reprotoxic (CMR) substances in categories 1A, 1B or 2 in toys or structurally separate parts, except if the substance is inaccessible or present in concentrations below a certain threshold. Plastic materials have been suggested as a relevant source of human exposure to endocrine disrupting chemicals (EDCs) [6–8].

Today, even if a common definition of what is an endocrine disrupter has been adopted at the European level, there is no regulatory obligations for those placing toys on the market to address this concern for endocrine effects yet. On the contrary, in the recently published Medical Device Regulation 2017/745, it is stated that substances having endocrine-disrupting properties with scientific evidence of probably serious effects on humans, shall only be present in a medical device above 0.1% by weight (w/w) when justified.

Phthalates are a group of substances, which have been widely used in toys as plasticizers to increase their flexibility. Several phthalates have been classified as reprotoxicants 1B and therefore been banned in toys or other articles which may be put into the children mouth. For example, the use of di(2-ethylhexyl) phthalate (DEHP), di-n-butylphthalate (DnBP), di-iso-butylphthalate (DiBP) and butylbenzyl-phthalate (BBP) is restricted to a maximum level of 0.1% in all toy plastic parts following entry 51 of Annex XVII to REACH [9]. In addition, the same maximum threshold has to be applied to di-iso-nonyl-phthalate (DiNP), di-n-octyl phthalate (DnOP), di-iso-decyl phthalate (DiDP) when children can place toys in their mouth (entry 52 of Annex XVII to REACH). Some phthalates have also been recently identified as endocrine disrupters under the REACH regulation [10].

Actually, the described low regulatory limit of 0.1% is equivalent to a ban of phthalates in plastics, as a minimal level 10% phthalates is required to achieve the softening effect on PVC. As a result, efforts have been made by industry to substitute many of the reprotoxic phthalates with less potent substances, such as bis-2-ethylhexyl terephthalate (DEHTP) or acetyl tributyl citrate (ATBC), in toys. However, for many substitutes endocrine activity data is not available. Other compounds used in plastics such as bisphenol A (BPA) have also been recently identified as endocrine disrupters under the REACH regulation [10].

In addition to chemicals with known endocrine disrupting potential, toys may contain substances that have never been evaluated for their endocrine activity. Especially for non-intentionally-added substances, such as degradation products of contaminants, little is known on their potential endocrine effects. To be able to detect not only known or suspected endocrine disruptors, but also currently unknown substances with an endocrine activity, biodetection

systems can be used [11,12]. Berger et al. [6] have detected significant endocrine activity in two out of eleven tested plastic teethers using Yeast Estrogen and Yeast Androgen screen assays. The estrogenic and antiandrogenic activity of one of these teethers was linked to methyl- and propyl paraben. Antiandrogenic effects of toys and babyproducts could also be detected by Szczepańska et al [13] using the Yeast Androgen screen. Using comparable approaches, endocrine activity was reported for several plastic food contact materials ([4, 14–17], canine plastic toys [18] and water from plastic bottles [8].

Thus, the purpose of this study was to analyze selected toys for leaching of endocrine active substances potentially absorbed by toy-mouthing children. Several studies have already been carried out to assess the health risks associated with substance exposure originating from toys [19–26]. Most of these studies, however, assessed exposure scenarios based on the composition as well as migration or emission tests to identify hazardous chemicals that need to be avoided or reduced in the material composition.

Hence, in the present study, toys and children's equipment intended to be used by children up to three years were screened for EDCs using effect-directed analysis [27]. The CALUX® assays (Chemically Activated Luciferase gene expression) were applied to detect substances (i.e. ligands) interacting with the human estrogen or androgen receptors and thereby modifying the subsequent transcriptional response [16, 28].

In a previous study, composition and migration tests were already conducted on these items, in order to search for plasticizers (phthalates and their substitutes) [5]. Out of 31 toys and items of children's equipment tested, only the PVC toys contained plasticizers, except for one elastomer toy containing ATBC. The plasticizers found included two prohibited phthalates (DEHP used as a principal plasticizer associated with DINP in a toy purchased at a street market, and low levels of DEHP attributed to its presence as an impurity of DEHTP in three toys), and some of their substitutes such as ATBC, DEHTP, diisononyl cyclohexane-1,2-dicarboxylate (DINCH), Di-2-ethylhexyl isophthalate (DOIP) or 2,2,4-Trimethyl-1,3-pentanediol diisobutyrate (TXIB) were quantified. Subsequently, it was shown that all the substances found migrated in a saliva simulant.

The current work focuses on the analysis of toy migrates using bioassays and known or suspected hormone active substances were screened by GC-MS and HPLC-MS/MS. The potential hormonal activities of these toys have been assessed.

## Materials and methods

### Selection of toys

A literature review focusing on extensive reports and surveys of professionals from the toy sector was performed to identify the materials and compounds likely to be most commonly used in toys in France and Europe [5]. The categories of toys sold most widely in the EU are pre-school and infant toys, followed by dolls, outdoor toys and board games/puzzles (more than half of all sales in the EU).

Toys and children's equipment represent a wide diversity of types. Their materials (hard/soft plastic, wood, textiles, metals, *etc*.) and the substances they are composed of vary in a great extent. Without this being an exhaustive list, they include the following substances: phthalates and substitutes, flame retardants, short-chain chlorinated paraffins, bisphenol A, metals, *etc*. In France, the best-selling toys over all age groups are construction toys, dolls, board games and puzzles, infant toys, teething rings and outdoor toys. Taking into account only toys made of plastic and the target population (children up to 36 months), this work focused on the following toy categories: infant toys, dolls and construction toys.

In a previous study, a set of toys has been analyzed for its concentrations of several endocrine active substances: tests were carried out in France to characterize toys present on the French market for which leaching of hazardous compounds may expose young children. Tests of the composition of phthalates and their substitutes in a limited sample of toys and children's equipment (bibs, teething rings, pacifiers), followed by migration testing in a saliva simulant were conducted on ANSES's request [5]. These tests were only conducted on new toys. Based on the results of these tests, 18 toys were selected by Anses for further testing to assess endocrine activity of the migration fluid. Table 1 shows the samples that were selected for analysis and ordered online.

## Characterization of materials

Toy samples were characterized following a 3-step approach. First, the toy itself, its packaging and the producers' web sites were examined for available product information (e.g. recycling codes). If not enough information could be obtained, a formal request asking for the used raw materials was directly sent to the toy manufacturer as a second step. Several toy producers agreed to share this information (see Table 1). In a third step the remaining samples were analyzed by Fourier-transform infrared (FT-IR) spectroscopy analysis using a FT-IR-spectrometer type Perkin-Elmer, model Spectrum One to identify the materials. As a general approach the samples were analyzed on the surface as well as in the internal part. Recordings were made using a universal ATR unit (equipped with Diamond/ZnSe crystal) in a spectral range from 4000–600 cm$^{-1}$.

## Migration experiments

To simulate the process of toy mouthing by children, the toy subparts were incubated with saliva simulant (0.82 mM $MgCl_2$ (Sigma), 1 mM $CaCl_2$ (VWR), 3.3 mM $K_2HPO_4$ (Sigma), 3.8 mM $K_2CO_3$ (Sigma), 5.6 mM NaCl (Sigma), 10 mM KCl (Sigma), pH of 6.8 –stored in the dark for 2 weeks). Migration experiments were conducted based on the JRC Scientific and Technical Report JRC EUR 23813 EN (2009) [29]. If possible, toys were migrated without any further cutting in a wide-neck 1000 ml glass bottle closed tightly with a polytetrafluoroethylene (PTFE) cap applying saliva simulant in a ratio of 100 ml per 10 cm$^2$ sample surface. Samples that did not fit into wide-neck 1000 ml glass bottles, were cut into pieces of 20 to 47 cm$^2$ adjusting the volume of saliva simulant accordingly to obtain a ratio of 100 ml per 10 cm$^2$. Some toys were split into different subsamples and analyzed independently as they were composed of different materials. For instance, a doll was split into three subsamples: dress, hair and face. For comparison, saliva blanks, filled in glass bottles with PTFE coated caps, were analyzed to check for possible contaminations of the saliva simulant. During migration, bottles were rotated on a head over heels rotator at 60 rpm at room temperature. After 30 min of rotation, the simulant was collected and replaced by fresh saliva again incubating with the sample for 30 min. Then, both migration batches were collected and pooled reaching a total volume of 400 ml. Simulants were stored at 4 to 8°C.

After migration, SPE with Oasis HLB columns (6 cc/200 mg) was carried out to concentrate the obtained migrates. The preparation procedure of the columns included conditioning with 15 ml of an ACN-MeOH 1:1 mixture and 5 ml MeOH, followed by equilibration with 5 ml ultrapure water. Thereafter the samples were applied to SPE using vacuum (around 0.7 bar), each column was washed with 5 ml ultrapure water and eluted with 4.5 ml of an ACN-MeOH 1:1 mixture. 200 μl DMSO was added to each vial as a "keeper", before the samples were eluted and evaporated to a final volume of 250–300 μl at vacuum (between -0.2 and 0.7 bar) and stored at 4°C prior to analysis by the bioassays.

Table 1. Samples selected for analysis and material characterization.

| Sample code | Game | Age class | Sampling area | Material |
|---|---|---|---|---|
| T01 | Bath book | From 4 months | Plastic part of page | EVA [b] |
| T02 | Plastic toy | 12 months—6 years | Plastic | PVC [a] |
| T03a | Book early age | Since birth | Fabric | Fabric |
| T03b | | | Battery lid | ABS [a] |
| T04a | First age toy | From 6 months | Support | Styrene acrylate ; core: PP/PE [a] |
| T04b | | | Orange ring | EVA [a] |
| T04c | | | Yellow ring | EVA [a] |
| T05a | Bath game | unspecified | Star | PVC [a] |
| T05b | | | Hippo | PVC [a] |
| T06a | Teething key ring | From 3 months | Blue key | PE [a] |
| T06b | | | Ring | PP-PE co-polymer [a] |
| T07a | Doll | From 4 years | Dress | Fabric |
| T07b | | | Hairs | Unknown |
| T07c | | | Face | PVC [a] |
| T08a | Doll | From 3 years | Arm | PVC [a] |
| T08b | | | Battery lid | ABS [a] |
| T08c | | | Fabric | Fabric |
| T09a | Doll | From18 months | Arm | PVC [a] |
| T09b | | | Fabric | Fabric |
| T10a | Building game | From18 months | Red brick | ABS [b] |
| T10b | | | Yellow brick | ABS [b] |
| T11a | Building game | 12 months– 5 years | Yellow brick | ABS [b] with sticker |
| T11b | | | Green brick | ABS [b] |
| T12a | Building game | From18 months | Cone | Wood |
| T12b | | | Triangle | Wood |
| T13a | Figure | From 18 months | Tailgate | ABS/steel [b] |
| T13b | | | Wheel | ABS/steel [b] |
| T14a | Outdoor play equipment | From 18 months | Ball | Polyurethane coating–PVC [a] |
| T14b | | | Bucket | PP [b] |
| T14c | | | Shovel | PP [b] |
| T15 | Ball | From 6 months | Soft shell | Polyurethane coating–PVC [a] |
| T16 | Soft toy | unspecified | Fabric | Fabric |
| T17 | Soft toy | Since birth | Fabric | Fabric |
| T18 | Soft toy | Since birth | Fabric | Fabric |

All sample treatment procedures have been previously validated for recovery and reproducibility using 10% ethanol migrates [16]. To demonstrate that the developed methods are suitable for the concentration of saliva solvent, recoveries and reproducibility were determined as previously described by Mertl et al. [16]. 1 liter of saliva solvent was spiked with 2,4-dihydroxy-benzophenone, bisphenol A, benzophenone and benzylbutylphthalate at a concentration of 10 ppb and extracted by SPE and transferred to DMSO. Concentrations of the spiked substances were determined by HPLC-UV/VIS in the concentrated DMSO extracts.

## Analysis of toys for hormonal activity with different *in vitro* tests

The DMSO extracts were analyzed by two different CALUX® bioassays (Estrogen Receptor α-CALUX® and Androgen receptor-CALUX®). CALUX® (Chemical Activated Luciferase

Expression) assays are human cell-based reporter gene assays for the detection of hormonal activity. In more detail, these assays measure estrogen/androgen and anti-estrogen/anti-androgen activity as previously described by Mertl et al. [16]. Both bioassays are based on a human U2-OS osteosarcoma cell line and show a highly sensitive and selective response to natural and synthetic estrogen- or androgen agonists. In the modified U2-OS cells, an activated steroid hormone receptor binds to the promoter region of a firefly luciferase reporter gene and activates firefly luciferase transcription. The firefly luciferase reporter gene leads to a light emission if luciferin and co-factors are added and can be very sensitively and selectively quantified. This light signal increases depending on the dose of estrogen or androgen-active substances [30].

Three independent migrates of each sample extract were analyzed in triplicates in both CALUX®-assays. The concentrated DMSO extracts were added to a final concentration of 0.5% to the assay medium. The activity was quantified by comparison to a linear regression curve of a dilution series of the natural estrogen 17β-estradiol (E2) or androgen 5α-dihydrotestosterone (DHT). Estrogen activities are expressed as 17β-estradiol equivalents (EEQ) and androgen activity are expressed as 5α-dihydrotestosterone (DHT) equivalents (AEQ) Activity of anti-estrogen activity is quantified by comparison to a dilution curve of the pharmaceutical estrogen antagonist 4-ortho-hydroxy-tamoxifen (4-OHT) and expressed as 4-OHT equivalents. Activity of anti-androgen activity is quantified by comparison to the androgen antagonist flutamide and is expressed as flutamide equivalents. In addition, all samples were further tested for cytotoxic effects or growth inhibiting effects by microscopy.

To test for antagonisms or inhibiting effects by the sample matrix, each sample was spiked with a non-saturating concentration of a suitable positive control (8 pmol/l of 17β-estradiol or 400 pmol/l of DHT) and analyzed via the respective CALUX® assay. If more than 40% of the activity of the spiked hormone was suppressed by the tested sample, the samples have to be diluted prior to analysis for hormone agonists, resulting in increased detection limits. Samples, which show such a significant suppression of the hormone activity of the spiked hormone, were categorized as antagonistic, if cytotoxic effects cannot explain the reduction of the activity.

Reproducibility of the CALUX® assays and robustness towards polymer extracts was determined in a previous study [16].

## Chemical trace analysis

The toys were further tested to identify which compounds may be responsible for recorded hormonal activities. Chemical trace analysis (GC-MS and HPLC-MS/MS) was performed on 34 toys or toy parts. Sample migrates were analyzed for 41 known or suspected endocrine active substances as well as common alternatives to endocrine active plasticizers. The selection of the 41 target substances was mostly based on a literature reviews by J. Muncke [7,31] on known and suspected endocrine active substances with authorized use in food packaging materials. The list was extended by further by some endocrine active substances, which are not authorized for food contact, but which are commonly present in plastics as degradation non-intentionally added substances: The styrene oligomers 1,3-Diphenylpropane and trans-1,2-Diphenylcyclobutane, which are estrogen active according to Kitamura et al. [32] and 2,4-di-tert-butylphenol and 2,4-dicumylphenol, which are degradation products of common antioxidants that are reported to be endocrine active [16,33]. In addition to already known or suspected endocrine active substances, the method was extended by plasticizers which are used as alternatives to endocrine active phthalates: Tributyl citrate (TBC), Bis(2-ethylhexyl)

adipate (DEHA), bis-2-propylheptyl phthalate (DPHP), Di-2-ethylhexylfumarat, ATBC, DOIP and DEHTP.

Gas chromatography analysis was carried out as previously described in Mertl et al. [16] extending the substance set for additional target substances. In short, gas chromatography analysis was carried out using the SBSE technique (Stir Bar Sorptive Extraction using the PDMS GERSTEL Twister) in combination with thermodesorption und GC-MS. 5 ml each sample were extracted on a magnetic stirring plate at 2000 rpm for 1 hour. After the extraction the Twister was remove, dried with a lint-free tissue paper and placed in a thermal desorption tube. The analyses were performed using a 6890N GC equipped with a 5975C (inert XL Triple Axis) Mass Selective Detector (Agilent Technologies, Waldbronn, Germany), a Thermal Desorption Unit (TDU), Cooled Injection System (CIS4) and Multi-Purpose Sampler (MPS) (Gerstel, Mühlheim an der Ruhr, Germany). The analytical conditions for the thermodesorption were: TDU in splitless mode, the temperature program started at 40˚C which was increased at a rate of 500˚C/min until 280˚C for 5 min; the CIS was cooled at -150˚C during the thermodesorption and then the temperature was increased at a rate of 12˚C/s until 300˚C for 3 min. The GC system was equipped with a HP-5MS (Agilent Technologies, Waldbronn, Germany) capillary column (30m x 250μm x 0,25μm). The GC temperature program ranged from 40˚C to 300˚C with a rate of 10˚C/min. The carried gas was helium at 1.2ml/min. Ionization was performed by electron impact mode (EI) at 70eV, the temperature of the transfer line was 300˚C. Identification and quantification of 28 target compounds as already described in Mertl et al. [16] adding 10 additional substances was done applying selected ion monitoring (SIM) mode. For each reference substance a calibration curve was determined for concentrations between 1 μg/l and 50 μg/l and LODs and LOQs were evaluated according to DIN 32 645. For the identification of unknown compounds SCAN mode using the NIST02 library (US National Institute of Standards and Technology) followed by comparison with known mass spectra was conducted. Two independent migrates were separately analyzed in duplicates. Conditioning of the twisters at 300˚C for ten minutes was done to avoid contaminations from previously tested samples.

Besides GC-MS analysis, concentrations of bisphenol A, F and S were measured with high-pressure liquid chromatography combined with tandem mass spectrometry (Dionex U3000 HPLC, AB-Sciex Qtrap 5500), using ESI (electrospray ionization) for ionization. An aliquot of 10 μl sample was injected on an ACE 3 C18-AR 150 x 3 mm (V13-7639) column. Mobile phase solvents were ultrapure water and acetonitrile in an initial ratio of 90:10. Separation was achieved at 50˚C using a flow rate of 600 μl/min with a linear increase of the initial acetonitrile concentration up to a concentration of 100% acetonitrile during the first 20 minutes and using 100% acetonitrile as mobile phase for 10 further minutes. Measurements with a negative multi-reaction-monitoring (MRM)-mode show two different stable transitions (parent ion–fragment ion) for each analyte, which were used for qualification and quantification, respectively. For quantification, peak areas were integrated using Analyst 1.6 and an external calibration for concentrations between 1 and 10 μg/l. The limits of detection and quantification were determined statistically according to DIN 32465. Precision was determined using the repeatability between five replicates of saliva solvent matrix spiked with a concentration of 10 μg/l of each substance and reported as a relative standard deviation (RSD). Accuracy was calculated as the percentage of recovery from three replicates at the same concentration of 10 μg/l.

Fig 1 summarizes how the endocrine potential of these toys has been investigated.

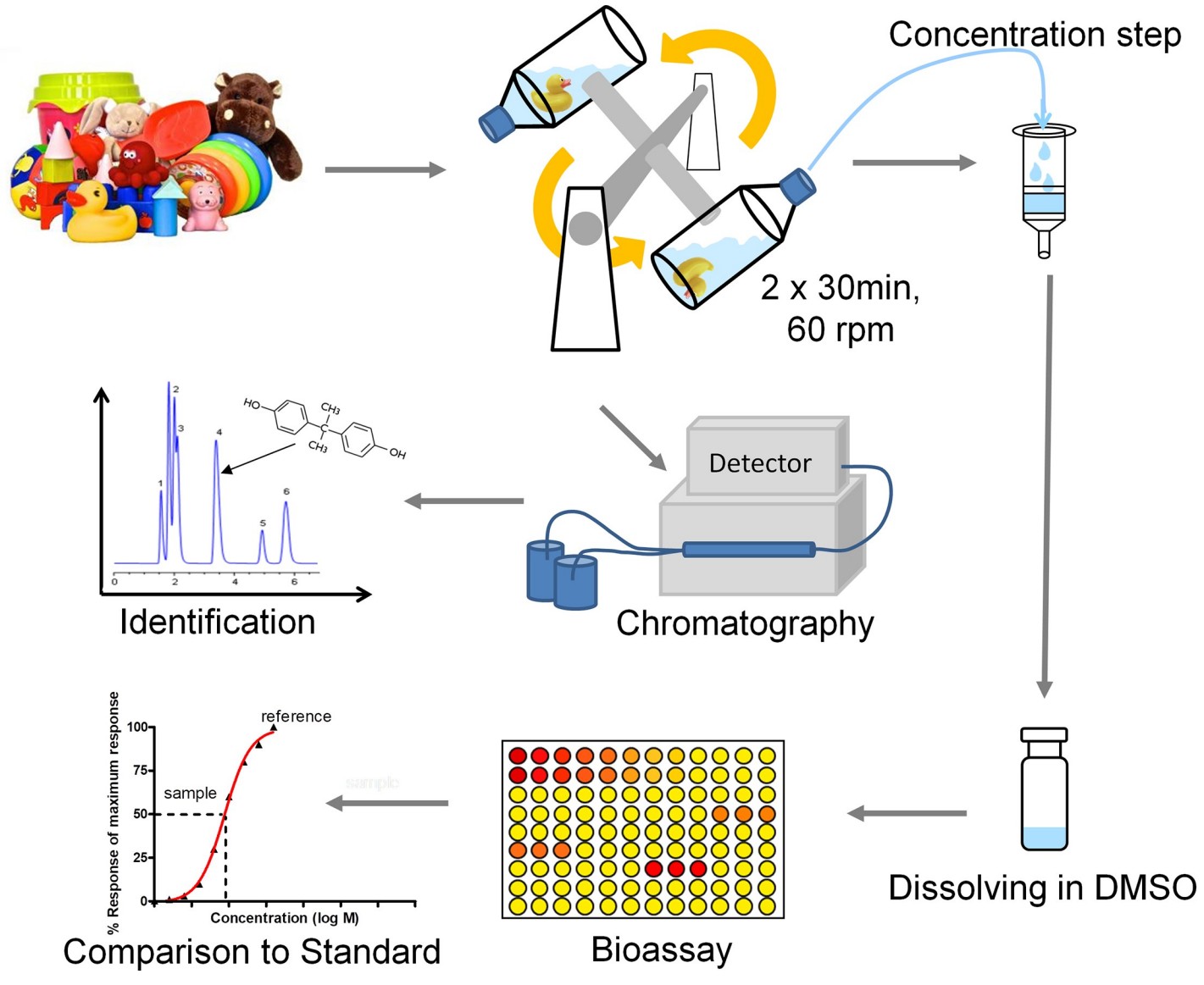

**Fig 1. Scheme of sample analysis.**

### Analysis of detected pure substances for endocrine activity

Substances detected by chemical target analysis (TBC, benzophenone and diethyl phthalate) in at least one sample were used to prepare a dilution series (100 mM, 30 mM, 10 mM, 3 mM, 1 mM, 0.3 mM and 0.1 mM) in DMSO, subsequently tested for endocrine activity in the ER CALUX® assay. Testing procedures followed instructions given in 2.4.

### Assessment of the health risks associated with the mouthing of plastic toys containing phthalate substitutes

Health risk assessment was performed for critical substances detected in the selected toys—the two phthalate substitutes acetyl-tributyl-citrate (ATBC, n° CAS 77-90-7), and diethylhexyl-terephthalate (DEHTP, n° CAS 6422-86-2) as well as for other potential active substances such

as tributyl citrate (TBC, n° CAS 77-94-1), bisphenol A (CAS 80-05-7) and benzophenone (CAS 119-61-9).

Exposure scenarios adapted from RIVM [20] were applied to estimate oral exposure from mouthing behavior of toys in children (0 to 12 months).

The following equation was used to calculate the daily exposure (DE) to the substances:

$$DE = F \text{ x } S \text{ x } D/BW \tag{1}$$

With:

- BW : body weight [kg]; a body weight of 8kg was considered

- F : migration flux [µg/min/cm$^2$]: values measured in this study for each compounds were taken

- S : surface in contact with the mouth [cm$^2$]; a surface of 10 cm2 was considered

- D : contact duration in the mouth [min/d] ; a duration of 180 min per day was considered

Risks were estimated by comparing the daily exposure (DE) to toxicological reference values (TRV) when available.

## Results and discussion

### Selected toys and characterization of materials

In a 3-step approach (research, contact to the manufacturer and FT-IR analysis) the materials of the tested toys were characterized. Table 1 gives an overview on all selected toys and sampling areas and lists the determined material of all tested sample parts.

### Endocrine activity of the toys migrates

To test the endocrine potential of selected toys, saliva migrates were prepared from each sample. Prior to bioassay analysis migrates have been concentrated. A suitable solid phase extraction pre-concentration method for 10% ethanol migrates was already validated for recovery and reproducibility by Mertl et al. [16]. The robustness of the method to dissolved polymers has already been demonstrated in the previous evaluation by Mertl et al. [16] on plastic samples extracted at much harsher conditions (10 days, 60°C, ethanol) which can be considered as a worst case. To verify the method applicability on saliva migrates, additional validation experiments were performed with saliva simulant. Therefore, saliva simulant was spiked with 4 representative reference substances to determine substance losses during SPE concentration. Targeted HPLC-UV/VIS analysis determined reasonable compound recoveries (benzophenone: 61 ± 5%, benzylbutylphthalate: 92 ± 4%, bisphenol A: 83 ± 10 and 2,4-dihydroxybenzophenone: 83 ± 3%).

After simulating mouthing by a child, the saliva simulant was concentrated by solid phase extraction, transferred to DMSO and analyzed using CALUX® bioassays. These biological assays can detect substances that can activate or inhibit the human estrogen or androgen receptor, including currently unknown hormone active substances.

The results of the bioassay screening are listed in Table 2. Nine out of 18 tested toys showed a significant estrogenic activity in at least one of the tested parts. The detected estrogenic activities ranged from 5.3 to 83 pg estradiol equivalents (EEQ)/cm$^2$ sample surface.

To evaluate if the sample matrix of the toy migrates has an inhibiting effect on the biodetection and to check for hormone antagonists, all samples were spiked with the natural hormones in non-saturating concentrations. Test responses of spiked sample migrates were compared to

**Table 2.** *Estrogenic and androgenic activity* of the selected toys (mean ± standard deviation of the analysis of three independent samples).

| Sample code | ER-CALUX® | | AR-CALUX® | |
| --- | --- | --- | --- | --- |
| | Estrogenic activity [pg EEQ/cm$^2$] | Anti-estrogen activity [pg 4-OHT equivalents/cm$^2$] | Androgen activity (pg AEQ/cm$^2$) | Anti-androgen activity [µg flutamide equivalents/cm$^2$] |
| T01 | < 4 | < 300 | < 120 | <0.3 |
| T02 | 83 ± 7 | < 300 | < 120 | <0.3 |
| T03a | < 4 | < 300 | < 120 | <0.3 |
| T03b | < 4 | < 300 | < 120 | <0.3 |
| T04a | < 4 | < 300 | < 120 | <0.3 |
| T04b | < 4 | < 300 | < 120 | <0.3 |
| T04c | < 4 | < 300 | < 120 | <0.3 |
| T05a | 54 ± 27 | < 300 | < 120 | <0.3 |
| T05b | 67 ± 22 | < 300 | < 120 | <0.3 |
| T06a | < 4 | < 300 | < 120 | <0.3 |
| T06b | < 4 | < 300 | < 120 | <0.3 |
| T07a | < 4 | < 300 | < 120 | <0.3 |
| T07b | < 4 | < 300 | < 120 | <0.3 |
| T07c | < 4 | < 300 | < 120 | <0.3 |
| T08a | < 4 | < 300 | < 120 | <0.3 |
| T08b | < 4 | < 300 | < 120 | <0.3 |
| T08c | 22 ± 10 | < 300 | < 120 | <0.3 |
| T09a | < 4 | < 300 | < 120 | <0.3 |
| T09b | 14 ± 7 | < 300 | < 120 | <0.3 |
| T10a | < 4 | < 300 | < 120 | <0.3 |
| T10b | < 4 | < 300 | < 120 | <0.3 |
| T11a | < 4 | < 300 | < 120 | <0.3 |
| T11b | < 4 | < 300 | < 120 | <0.3 |
| T12a | < 4 | < 300 | < 120 | <0.3 |
| T12b | < 4 | < 300 | < 120 | <0.3 |
| T13a | < 4 | < 300 | < 120 | <0.3 |
| T13b | < 4 | < 300 | < 120 | <0.3 |
| T14a | 2.6 ± 4.5 (1 of 3 extracts positive) | < 300 | < 120 | <0.3 |
| T14b | < 4 | < 300 | < 120 | <0.3 |
| T14c | < 4 | < 300 | < 120 | <0.3 |
| T15 | < 4 | < 300 | < 120 | <0.3 |
| T16 | 6.6 ± 2.2 | < 300 | < 120 | <0.3 |
| T17 | 5.3 ± 1.8 | < 300 | < 120 | <0.3 |
| T18 | 19 ± 17 | < 300 | < 120 | <0.3 |

No androgenic activity could be detected in any of the tested samples.

spiked solvent blanks. All tested samples showed the expected response to the natural hormones—no significant inhibition of the estrogenic or androgenic activity was observed. This demonstrates, that the test worked properly in the presence of the sample matrix and shows no indication for estrogen or androgen antagonists in the samples.

The PVC sample T02 showed the highest estrogenic activity of all tested samples (83 ± 7 pg EEQ/cm$^2$). An activity of 83 pg EEQ/cm$^2$ means that substances migrating from one square centimeter toy surface have the same estrogenic activity in the bioassay, as 83 pg of the natural female sex hormone 17β-estradiol. If plastic components or other industrial chemicals were

responsible for the determined estrogen activity, a much higher concentration would be needed to explain the reported estrogenic activity, as estrogen mimicking substances are generally not fitting as well into the receptor-binding site as the hormone itself [30].

Based on the assumptions by RIVM [20] (surface of 10 cm$^2$ for 3 hours in contact with saliva) 2.5 ng estradiol equivalents were calculated as a maximum daily uptake for the sample with the highest estrogen activity (calculated from 0.083 ng EEQ/cm$^2$ detected in a 60 minute extraction to 2.5 ng EEQ in 10 cm$^2$ in 180 minute extraction time).

In comparison to estrogen activities taken up by food, the detected estrogen activities were low. Many food products naturally contain estrogens, for instance 23 ng/L of the natural female sex hormone 17β estradiol were analytically determined in cow milk by Courant et al. [34]. By orders of magnitude higher estrogen activities than in cow milk, can be found in soy bean rich food, that contains phytoestrogens [35]. The maximum total daily uptake of estrogen activity by food is estimated to be up to 10,000 ng estradiol equivalents per day for babies who are fed by milk on soy bean basis [35]. The activities detected in the migrates of the saliva solvent were also significantly lower than estrogen activities previously reported for mineral water [7, 36, 37], where no estrogen activity would be expected. That said, comparisons between *in vitro* estrogen activities in food and toys should be interpreted with caution, because the potential health effects depend highly on the substance causing the activity. Many aspects of endocrine action in the human body, like metabolization and half-life are not taken into account in bioassays. Very importantly, some man-made estrogens have much higher half-lives in the human body. A prominent example is the contraceptive 17α ethinyl estradiol. It has approximatively the same *in vitro* estrogen activity as the natural sex hormone 17β-estradiol [38], but has, by orders of magnitude, higher activity in the human body, if taken up orally [39]. This is also reflected in the acceptable daily intake values calculated for these substances. For the natural estrogen 17β-estradiol the World Health Organization calculated an acceptable daily intake (ADI) of 3 μg per day for an adult person. Referring to a child's body weight of 8 kg, 2.5 ng estradiol equivalents, which was calculated as the maximum uptake by the worst-case toy, would still be well below the ADI for the natural hormone 17β estradiol. For 17α-ethinyl estradiol, however, an approximately 400-fold lower ADI of 7 ng for an adult person (60 kg) was calculated by Caldwell, *et al.* [40] based on working place thresholds. Therefore, for the synthetic estrogen 17α-ethinyl estradiol, the same activity would already be clearly above the calculated ADI for a child of 8 kg bodyweight.

Although the activity detected in the sample migrates was most probably neither caused by natural estrogens nor by 17α ethinyl estradiol these examples were described to illustrate that the nature of the responsible substance is influencing the hazard potential of an endocrine active sample. Consequently, no risk assessment is possible based on *in vitro* estrogen activities alone. Therefore, a chemical analysis was conducted in parallel to the *in vitro* assays.

## Chemical trace analysis

To further investigate the substances causing the reported estrogenic activities in toy migrates, migrates were analyzed by chemical trace analysis.

GC-MS and HPLC-MS/MS tsarget screening. The validation parameters, including the correlation coefficient (r) from the linear regression curves are listed in Table 3.

Table 4 lists the result from screenings for 41 target compounds by GC-MS and HPLC-MS/MS. Among them, six substances could be detected in at least one of the tested samples—that are: diethyl phthalate, benzophenone, TBC, ATBC, BPA and DEHTP. The remaining 33 substances could not be detected in any of the tested samples.

**Table 3. Validation parameters for GC-MS and HPLC-MS target analysis.**

| Substance | CAS -Number | LOD [µg/l] | LOQ [µg/l] | Recovery [%] | RSD [%] | r | Method |
|---|---|---|---|---|---|---|---|
| 1,4-Dichlorobenzene | 106-46-7 | 0.5 | 1.8 | 98% | 9.4 | 0.9997 | GC-MS |
| 4-chloro-3-methyl-phenol | 59-50-7 | 1.0 | 3.6 | 95% | 2.6 | 0.9990 | |
| Methyl p-hydroxybenzoate (Methylparaben) | 99-76-3 | 0.7 | 2.6 | 97% | 3.9 | 0.9975 | |
| Butylated hydroxyanisole (BHA) | 25013-16-5 | 0.9 | 3.2 | 98% | 5.9 | 0.9994 | |
| 2-Phenylphenol | 90-43-7 | 1.2 | 4.4 | 97% | 4.6 | 0.9986 | |
| Ethyl-4-hydroxy-benzoate (Ethylparaben) | 120-47-8 | 1.4 | 5.0 | 97% | 6.8 | 0.9981 | |
| Diethyl phthalate | 84-66-2 | 1.1 | 3.8 | 99% | 7.1 | 0.9944 | |
| n-Propyl-p-hydroxybenzoate (Propylparaben) | 94-13-3 | 0.8 | 2.9 | 100% | 5.2 | 0.9994 | |
| Benzophenone | 119-61-9 | 1.1 | 3.7 | 99% | 5.2 | 0.9989 | |
| 4-Phenylphenol | 92-69-3 | 0.8 | 2.7 | 100% | 6.7 | 0.9972 | |
| p-Cumyl phenol | 599-64-4 | 0.5 | 1.7 | 100% | 5.9 | 0.9989 | |
| 4-Nonylphenol (NP) | 104-40-5 | 1.1 | 4.0 | 98% | 2.6 | 0.9988 | |
| Dibutyl phthalate (DBP) | 84-74-2 | 0.8 | 2.9 | 104% | 2.8 | 0.9994 | |
| Oxybenzone | 131-57-7 | 0.5 | 1.7 | 99% | 4.4 | 0.9998 | |
| Triclosan | 3380-34-5 | 0.7 | 2.4 | 94% | 2.2 | 0.9978 | |
| 2,4-Dihydroxybenzophenone | 131-56-6 | 0.5 | 1.6 | 102% | 7.0 | 0.9991 | |
| 2,20-Dihydroxy-4-methoxybenzophenone | 131-53-3 | 0.7 | 2.3 | 92% | 4.0 | 0.9981 | |
| Di-n-hexylphthalate (DnHP) | 84-75-3 | 0.5 | 1.9 | 103% | 4.7 | 0.9987 | |
| Butyl benzyl phthalate (BBP) | 85-68-7 | 0.5 | 1.8 | 100% | 6.3 | 0.9988 | |
| Oleamide | 301-02-0 | 1.0 | 3.3 | 94% | 3.7 | 0.9958 | |
| 4,4´-Thiobis(6-terc-butyl-3-methyl-phenol) | 96-69-5 | 0.6 | 1.9 | 97% | 9.1 | 0.9986 | |
| 2,2´-Methylene bis(4-methyl-6-tert-butylphenol) | 119-47-1 | 0.6 | 2.1 | 102% | 5.6 | 0.9983 | |
| 2,2´-Methylenebis(4-ethyl-6-tert-butylphenol) | 88-24-4 | 0.6 | 2.2 | 101% | 5.1 | 0.9982 | |
| Dicyclohexyl phthalate | 84-61-7 | 0.6 | 2.0 | 102% | 4.7 | 0.9986 | |
| Bis (2-ethylhexyl) phtalate (DEHP) | 117-81-7 | 0.3 | 1.1 | 98% | 4.3 | 0.9996 | |
| Diphenyl-p-phenylenediamine | 74-31-7 | 0.5 | 1.8 | 97% | 5.4 | 0.9988 | |
| 4-Methylbenzophenone | 134-84-9 | 0.8 | 3.0 | 98% | 3.7 | 0.9993 | |
| 1,3-Diphenylpropane | 1081-75-0 | 1.1 | 3.9 | 101% | 5.5 | 0.9989 | |
| trans-1,2-Diphenylcyclobutane | 20071-09-4 | 1.1 | 3.9 | 99% | 5.5 | 0.9989 | |
| 2,4-di-tert-Butylphenol | 96-76-4 | 1.0 | 3.7 | 104% | 2.5 | 0.9990 | |
| 2,4-Dicumylphenol | 2772-45-4 | 0.5 | 1.7 | 102% | 2.9 | 0.9989 | |
| Tributyl citrate (TBC) | 77-94-1 | 1.2 | 4.4 | 95% | 4.9 | 0.9986 | |
| Bis(2-ethylhexyl) adipate (DEHA) | 103-23-1 | 0.6 | 2.1 | 100% | 4.4 | 0.9984 | |
| (bis-2-propylheptyl phthalate) DPHP | 53306-54-0 | 0.6 | 2.1 | 95% | 9.9 | 0.9983 | |
| Di-2-ethylhexylfumarat | 141-02-6 | 1.5 | 5.0 | 95% | 5.1 | 0.9897 | |
| Acetyl tributyl citrate (ATBC) | 77-90-7 | 0.9 | 3.1 | 87% | 14.3 | 0.9964 | |
| Bis-2-ethylhexyl isophthalate (DOIP) | 137-89-3 | 0.8 | 2.7 | 97% | 9.2 | 0.9973 | |
| DEHTP (Bis-2-ethylhexyl terephtalate ) | 6422-86-2 | 0.8 | 2.7 | 95% | 9.2 | 0.9973 | |
| Bisphenol A | 80-05-7 | 0.4 | 1.6 | 100% | 1.1% | 1.0000 | HPLC-MS/MS |
| Bisphenol F | 620-92-8 | 2.2 | 8.0 | 101% | 1.5% | 0.9998 | |
| Bisphenol S | 80-09-1 | 1.4 | 5.2 | 103% | 3.2% | 0.9995 | |

LOD: limit of detection

LOQ: limit of quantification

R: correlation coefficient of linear regression curve

RSD: relative standard deviation of recovery from 6 replicates

**Table 4. Substances detected in HPLC-MS/MS and GC-MS screenings for 41 target compounds (mean value ± standard deviation of 3 independent sample migrates in μg/cm² surface) and GC-MS non-target analysis.**

| Sample | Detected substance | CAS | concentration [μg/cm²] |
|---|---|---|---|
| T01 | Diethyl phthalate | 84-66-2 | 0.092 ± 0.011 |
| | TXIB [2] | 6846-50-0 | no quantification |
| T02 | Benzophenone | 119-61-9 | 0.33 ±0.02 |
| | DINCH [1] | 166412-78-8 | no quantification |
| | TXIB [1] | 6846-50-0 | no quantification |
| T03a | No substances identified | - | - |
| T03b | No substances identified | - | - |
| T04a | No substances identified | - | - |
| T04b | No substances identified | - | - |
| T04c | No substances identified | - | - |
| T05a | TBC | 77-94-1 | 3.7 ± 0.4 |
| | ATBC | 77-90-7 | 76 ± 23 |
| | BPA [3] | 80-05-7 | 1.3 ± 0.1 |
| | tributyl prop-1-ene-1,2,3—tricarboxylate [2] | 7568-58-3 | no quantification |
| | DINCH [1] | 166412-78-8 | no quantification |
| T05b | TBC | 77-94-1 | 3.0 ± 0.2 |
| | ATBC | 77-90-7 | 54 ± 12 |
| | BPA [3] | 80-05-7 | 1.1 ± 0.0 |
| | DINCH [1] | 166412-78-8 | no quantification |
| T06a | No substances identified | - | - |
| T06b | No substances identified | - | - |
| T07a | tributyl prop-1-ene-1,2,3—tricarboxylate [1] | 7568-58-3 | no quantification |
| T07b | BPA [3] | 80-05-7 | 1 out of 3 migrates:< LOQ, > LOD [2] |
| T07c | ATBC | 77-90-7 | 0.083 ± 0.029 |
| | Oxiranecarboxylic acid, 3-methyl-3-phenyl-, ethyl ester, (cis- and trans-) [2] | 19464-92-7&19464-95-0 | no quantification |
| | tributyl prop-1-ene-1,2,3—tricarboxylate [2] | 7568-58-3 | no quantification |
| | TXIB [1] | 6846-50-0 | no quantification |
| T08a | Benzophenone | 119-61-9 | 12 ± 2 |
| | TBC | 77-94-1 | 0.26 ± 0.05 |
| | ATBC | 77-90-7 | 18 ± 5 |
| | tributyl prop-1-ene-1,2,3—tricarboxylate [2] | 7568-58-3 | no quantification |
| | TXIB [1] | 6846-50-0 | no quantification |
| T08b | No substances identified | - | - |
| T08c | tributyl prop-1-ene-1,2,3—tricarboxylate [1] | 7568-58-3 | no quantification |
| | 3-hydroxy-2,2,4-trimethylpentyl isobutyrate [2] | 74367-34-3 | no quantification |
| T09a | Benzophenone | 119-61-9 | 0.57 ± 0.16 |
| | TBC | 77-94-1 | 2.2 ± 0.4 |
| | ATBC | 77-90-7 | 6.0 ± 2.3 |
| | TXIB [1] | 6846-50-0 | no quantification |
| | 3-hydroxy-2,2,4-trimethylpentyl isobutyrate [2] | 74367-34-3 | no quantification |
| | tributyl prop-1-ene-1,2,3—tricarboxylate [2] | 7568-58-3 | no quantification |
| T9b | Benzophenone | 119-61-9 | 0.38 ±0.14 |
| | TXIB [1] | 6846-50-0 | no quantification |
| T10a | No substances identified | - | - |
| T10b | No substances identified | - | - |
| T11a | Benzophenone | 119-61-9 | 0.36 ±0.21 |

*(Continued)*

**Table 4.** (Continued)

| Sample | Detected substance | CAS | concentration [µg/cm$^2$] |
|--------|-------------------|-----|--------------------------|
| T11b | No substances identified | - | - |
| T12a | Isobutyric acid 2-ethyl-3-hydroxyhexyl ester [2] | 74367-31-0 | no quantification |
| T12b | Isobutyric acid 2-ethyl-3-hydroxyhexyl ester [2] | 74367-31-0 | no quantification |
| T13a | No substances identified | - | - |
| T13b | 13-Docosenamide, (Z)-[1] | 112-84-5 | no quantification |
| T14a | DEHTP | 6422-86-2 | 12 ± 9 |
| T14b | No substances identified | - | - |
| T14c | No substances identified | - | - |
| T15 | DEHTP | 6422-86-2 | 1.0 ± 0.1 |
|  | BPA[3] | 80-05-7 | < LOQ |
|  | TXIB[1] | 6846-50-0 | no quantification |
| T16 | No substances identified | - | - |
| T17 | No substances identified | - | - |
| T18 | No substances identified | - | - |

(1) Substance, identified in GC-MS non-target-screening, identification verified by comparison to an external reference standard

(2) Substance, identified in GC-MS non-target-screening, no verification of the identification

(3) Substance, identified in HPLC-MS/MS

Two substances, of the six detectable ones, have known endocrine properties, namely, bisphenol A (BPA) and benzophenone [41,42]. BPA is a monomer used in the production of polycarbonate and epoxy resins and is an integral part of thermo ink paper. These materials are therefore considered as the main sources for exposure to BPA [43]. The two parts of toy "T05", in which BPA could be detected, consist of neither of these materials, but of flexible PVC. BPA has previously been a common additive in PVC as a production aid to stabilize vinyl chloride monomer, but the use of BPA as a stabilizer for vinyl chloride monomer was stopped in Europe since 2001. BPA has also been restricted in food contact plastic in Europe. Nevertheless, BPA still seems to be used in the PVC production for toys [43] outside the scope of the European plastics regulation (EU) No 10/2011. The detected BPA concentrations in the samples T05a and T05b were well reproducible for three independently prepared migrations. With approximately 1mg BPA migration per cm$^2$ sample surface, the migration is by orders of magnitude higher than migration from polycarbonate or epoxy materials to water or beverages, even though the migration time and temperature were significantly lower in this study [44]. Besides "T05", BPA could be detected in one out of three independently prepared migrates of sample T07b. However, the detected concentration was too low to be quantified.

Benzophenone was identified in five different tested sample parts (T02, T08a, T09a, T09b and T11a). Four of these are built from flexible PVC, one sample (T11a) is made of Acrylonitrile butadiene styrene (ABS) with a sticker of printed paper on it. In the latter one printing inks from the paper sticker seem to be the most plausible source for the detected benzophenone, as benzophenone is a common photo initiator in UV printing colors [45]. In PVC benzophenone is often us as a UV-stabilizer.

The 41 target compounds in the analysis include seven phthalates, previously widely used in plastic toys. Except for toy sample "T01", where trace amounts of diethylphthalate were detected, none of the 18 tested toys leached significant amounts of these phthalates. In toys made of flexible PVC, the common phthalate alternatives TBC, ATBC, DEHTP and DINCH were detected. When simulating mouthing, these plasticizers leached into saliva solvent in

concentrations up to 76 μg per cm$^2$ toy surface. While many phthalates are suspected to have endocrine disrupting potential, the detected alternatives are currently not considered as potential endocrine disruptors [10].

The tested samples were intended to reflect a realistic product status after delivery to the consumer. Therefore, some of the detected endocrine active substances might have been introduced by contaminations from packaging or during transport. Samples have indeed been ordered online and were mostly delivered in recycled cardboard boxes, which are known to frequently contain endocrine active substances, such as BPA, phthalates or benzophenone [46,47]. However, especially in the case of the relatively high BPA concentrations detected in sample T05, it does not seem plausible that the detected BPA originates from the cardboard. Migration experiments by Suciu et al. (2013) showed, that migration to dry food is low for BPA (<1%) even in direct contact, while higher migration could be expected from phthalates [48]. Phthalates are typically present at comparable concentrations in recycled cardboard but could not be detected in 33 out of 34 tested sample parts. Further, the migration was reproducibly detected in all 6 independent species of sample T05 (3 of part T05a, and 3 of part T05b), although not all sample parts had the same contact area to the cardboard. Therefore, the cardboard does not seem to be a likely source for the relatively high amounts of BPA detected in sample T05. Nevertheless, an additional analysis of packaging material and investigation of samples that are carefully drawn at production sites would be interesting for future investigations, to get better information on the source of the detected substances.

GC-MS Non-target screening. In addition to the target screening analysis for 41 substances, further substances could be identified in a GC-MS non-target screening by comparison of the mass spectra to the NIST 14 database (see Table 4).

The plasticizers DINCH and TXIB that are commonly used as phthalate alternatives could be identified in several PVC samples. Furthermore, *tributyl prop-1-ene-1,2,3—tricarboxylate* was identified in 6 samples. It is a less commonly used plasticizer and has been previously proposed as a safe alternative to phthalates [2].

*Isobutyric acid 2-ethyl-3-hydroxyhexyl ester* and *3-hydroxy-2,2,4-trimethylpentyl isobutyrate* were identified at the same retention time in different samples by comparison to NIST database. However, these identifications are assumed to be wrong assignments, as it seems more likely, that the detected peaks represent the TXIB hydrolysis product *(1-hydroxy-2,4,4-trimethylpentan-3-yl) 2-methylpropanoate*.

13-Docosenamide, detected in sample "T13b" and more commonly known as erucamide, is one of the most common slip agents in polyolefin and styrene polymerization. *13-docosenamide* is not endocrine active in the *in vitro* tests used in this study.

*Ethyl-2,3-epoxy-3-phenylbutyrate* was identified in T07c. This identification was not confirmed by comparison to a reference substance, but it seems plausible, as the identified substance is used as compound in strawberry fragrances [49], and the tested sample part had a strawberry odor.

As pure substances were only available for DINCH, TXIB and 13-Docosenamide, none of the other identifications could be confirmed by comparison to reference standards.

**Correlation between bioassays results and chemical trace analysis.** In order to identify the source of the hormonal activity in positively tested samples in the ER CALUX®, results from chemical analysis and bioassays were compared. For this purpose, all detected substances were tested as pure chemicals in the estrogen receptor CALUX®, if they were available. The response curves for the positively tested substances in the bioassay are shown in Fig 2. From that the half-maximal effective concentration (EC50) was calculated for all compounds showing estrogenic effects in the ER-CALUX® (see Table 5).

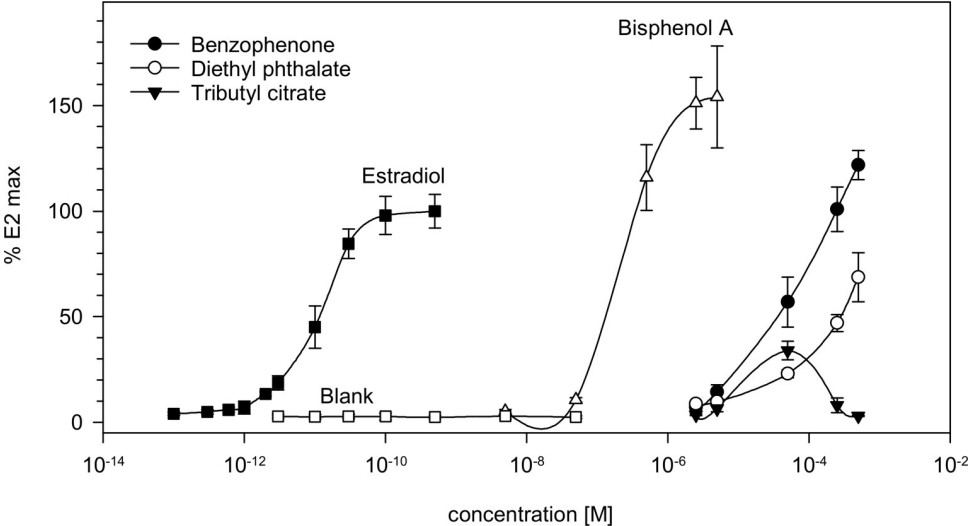

**Fig 2. Analysis of detected pure substances in ER-CALUX®: Response of bisphenol A (BPA), diethylphthalate (DEP), tributyl citrate (TBC) and benzophenone in comparison to natural estrogen 17β-estradiol and a solvent blank.** The y-axis ("E2 max") refers to the relative luciferase activity compared to the highest signal determined in the standard curve of 17β-estradiol (E2). The two highest concentrations of TBC led to a decrease in cell viability leading to decreased activities in the assay.

The determined estrogen activities of BPA, benzophenone and DEP were to be expected based on previously published results [2,41,42]. The phthalate alternatives TXIB, ATBC, DINCH and DEHTP were tested negative in the ER-CALUX®. Unexpectedly, the plasticizer TBC (Merck) showed a low, but significant estrogen activity at high concentrations, although the structure of the molecule would not suggest binding affinity to the estrogen receptor. It is possible, that this low estrogen activity, was caused by contaminants of this substance, rather than the substance itself, as the purity of the tested substance was below 97%.

For substances, that could not be obtained as pure chemicals a literature survey was conducted. tributyl prop-1-ene-1,2,3—tricarboxylate, Isobutyric acid 2-ethyl-3-hydroxyhexyl ester and 3-hydroxy-2,2,4-trimethylpentyl isobutyrate are not registered yet under the REACH regulation. No data on the endocrine activity of these substances is available.

By comparing the results from Table 2, Table 4 and Table 5 some, but not all determined estrogen activities can be explained by detected substances in the migrates.

**Table 5. EC$_{50}$ values of detected pure substances in ER-CALUX®.**

| Substance | CAS# | Estrogen active [yes/no] | EC$_{50}$ [mol/l] |
|---|---|---|---|
| Diethyl phthalate (DEP) | 84-66-2 | yes | $2*10^{-4}$ |
| 2,2,4-Trimethyl-1,3-pentanediol diisobutyrate (TXIB) | 6846-50-0 | no | $> 5*10^{-4}$ |
| 13-Docosenamide (Z) | 112-84-5 | no | $> 5*10^{-4}$ |
| Benzophenone | 119-61-9 | yes | $5*10^{-5}$ |
| Diisononyl cyclohexane-1,2-dicarboxylate (DINCH) | 166412-78-8 | no | $> 5*10^{-4}$ |
| Tributyl citrate (TBC) | 77-94-1 | yes | $5*10^{-5}$ |
| Acetyl tributyl citrate (ATBC) | 77-90-7 | no | $> 5*10^{-4}$ |
| Bisphenol A (BPA) | 80-05-7 | yes | $3*10^{-7}$ |
| DEHTP (Bis-2-ethylhexyl terephtalate) | 6422-86-2 | no | $> 5*10^{-4}$ |
| 17β-estradiol (positive control) | 50-28-2 | yes | $1*10^{-11}$ |

In the estrogen active migrates of samples T08c, T14a, T16, T17 and T18 no estrogen active substances could be identified by chemical analysis.

In migrates of the most estrogen active sample T02, the estrogen active UV-blocker benzophenone could be detected; however, the detected concentration only explains a small fraction of the detected estrogen activity. Based on the response curves for benzophenone and 17β estradiol in Fig 2, benzophenone is approximately by a factor of $5 \times 10^6$ less estrogen active then the natural estrogen 17β estradiol. Therefore, the detected concentration of 0.3 μg benzophenone/cm$^2$ explains less than 1% of the determined estrogen activity of the sample. Therefore, additional unknown substances have contributed to the estrogen activity.

The same seems true for sample T9b. T9b migrates show an estrogenic activity of 67 pg EEQ/cm$^2$. However, T9b contains benzophenone only in low amounts of 0.38 μg/cm$^2$. Considering the discussed activity differences between 17β estradiol and benzophenone again less than 1% of the measured activity can be explained by the reported benzophenone concentration in the sample migrate.

In the migrates of the two estrogen active samples T05a and T05b, the well-known estrogen active substance BPA could be detected. In the ER-CALUX$^{®}$, BPA is approximately by a factor of 30.000 less estrogen active than the natural estrogen 17ß-estradiol (see Fig 2). Based on this factor, the estrogen activity of 54 pg EEQ/cm$^2$ detected in sample T05a can be well explained by a BPA concentration of 1.3 μg/cm$^2$, taking into consideration the uncertainty of a biological test method. Applying the same logic, the activity of 38 pg EEQ/cm$^2$ detected in sample T05b matches a BPA concentration of 1.1 μg /cm$^2$ detected by target analysis. In addition to BPA, TBC was detected in both samples but based on the response curve for TBC, shown in Fig 2, the detected concentration would not significantly contribute to the total estrogen activity of the sample.

It was not always possible to link the observed endocrine activity with identified compounds. Either the chemicals in question is not detectable by the applied GC-MS method, or the substances are detected, but not covered by the NIST library. Moreover, among all possible endocrine active substances, analysis was performed for a limited set of substances by chemical analysis. Therefore, it is possible that the responsible substance cannot be detected. Thus, it would be very difficult to perform a quantitative risk assessment without an identification of the substance involved.

## Risk assessment for detected substances

The concentrations of the compounds following chemical trace analysis were analyzed in the light of their potential effects on children.

For benzophenone, based on assumptions by RIVM [20], a maximum daily exposure of 1.4 **μ**g/kg bodyweight can be calculated from the reported migration into the saliva solvent.

The calculation was done as described in the following:

i. sample T02: 0.33 μg/cm$^2$ in 60 min equivalent to 9.9 μg in 10 cm$^2$ in 180 min for a child of 8 kg bodyweight, a maximum daily uptake of 1.2 μg/kg.

ii. sample T09b: 0.38 μg/cm$^2$ in 60 min equivalent to 11.4 μg in 10 cm$^2$ in 180 min or for a child of 8 kg bodyweight, a maximum daily uptake of 1.4 μg/kg.

These estimated daily uptakes from toys, are still below the tolerable daily intake (TDI) of 30 μg/kg body weight per day, which was recommended by the European Food Safety Agency [43].

For DINCH, considering that the lowest reference value published for this compound is 0.7 mg/kg bw/day [50], the exposure *via* toys is not expected to raise any safety concern for children.

Based on published data, ATBC does not seem to show estrogenic or androgenic activity but there are doubts concerning activation of the PXR-receptor pathway which could affect the metabolism of steroid hormones [2]. Therefore, no conclusion can currently be drawn on the endocrine-disrupting nature of ATBC, because no robust data are available on potential effects such as those on thyroid function.

Concerning TBC, there is also very limited toxicological data available to assess the endocrine effects for human health but based on similar chemical characteristics, an analogue approach seems plausible for systemic effects [51]. Therefore, as for ATBC, no conclusion can currently be drawn on the endocrine-disrupting nature of TBC.

For BPA, the modes of action for each of the adverse effects are not well described yet but estrogenic or anti-androgenic activities are compatible with the reported effects [41]. A maximum daily exposure of 4,9 µg/ kg bw/day can be calculated from the detected concentration of BPA in the saliva solvent in the toys T05.

The calculation was done as described in the following:

i.  sample T05a: 1.3 µg/cm$^2$ in 60 min equivalent to 39 µg in 10 cm$^2$ in 180 min for a child of 8 kg bodyweight, a maximum daily uptake of 4.9 µg/kg.

ii. sample T05b: 1.1 µg/cm$^2$ in 60 min equivalent to 33 µg in 10 cm$^2$ in 180 min or for a child of 8 kg bodyweight, a maximum daily uptake of 4.1 µg/kg.

The daily uptake is just above the temporary tolerable daily intake (TDI) of 4 µg/kg body weight per day, which was recommended by the European Food Safety Agency (EFSA) in their re-evaluation of the risks of Bisphenol A taking into account endocrine effects [43].

## Conclusion

By applying different *in vitro* bioassays, we were able to detect significant estrogenic activity in at least one part of eight out of 18 tested toy items.

The tested samples were intended to reflect a realistic product status after delivery to the consumer. Therefore, it is possible that some of the detected substances do not originate from the toy itself, but were introduced during storage or transport, e.g. by contaminations from recycled cardboard.

Therefore, direct conclusions on the origin of estrogen active substances have to be drawn with care. To ensure that detected substances originate from the product, samples would have to be drawn directly at the production site and transported under controlled conditions, e.g. wrapped in aluminium foil. However, the samples tested in this study, are more representative for toys actually used by children than samples drawn directly at the factory.

In two of the tested samples, the well-known endocrine-active substance BPA could be identified as the main source of the detected estrogen activity. Based on the results, the exposure to BPA by mouthing of these toys could be above the temporary TDI, that EFSA has calculated for bisphenol A, demonstrating that some toys could significantly contribute to the total exposure to BPA of babies and infants.

For seven other estrogen active toy items (including the most estrogen active sample in the screening), the source of the estrogen activity could not be explained by chemical analysis for known or suspected endocrine active substances in plastics, indicating that the effect was caused by currently unknown endocrine active substances.

However, compared to the exposure to natural estrogens in food (e.g. phytoestrogens), the total uptake of estrogen activity by mouthing of toys seems to be low based on the results in this study.

## Supporting information

**S1 Table. Data from the analysis of detected pure substances in ER-CALUX® (data for Fig 2).**
(DOC)

## Acknowledgments

We would also like to thank Dr. Lidija Spoljaric-Lukacic for performing the FT-IR analysis of the toys, and Daniela Neubert for conducting HPLC-UV/VIS analysis. Furthermore, we would like to thank Elisa Mayrhofer for supporting with proof reading, graphs and revising the draft and Dr. Michael Pyerin, Dr. Manfred Tacker for supporting the research work with their expertise. We would further like to thank BDS Biodetection Systems, Amsterdam, for providing their CALUX cell line and supporting us with their knowledge.

## Author Contributions

**Conceptualization:** Aurelie Mathieu-Huart, Christophe Rousselle.

**Investigation:** Christian Kirchnawy, Fiona Hager, Veronica Osorio Piniella, Mathias Jeschko, Johannes Mertl.

**Methodology:** Christian Kirchnawy, Fiona Hager, Michael Washüttl.

**Project administration:** Michael Washüttl.

**Resources:** Michael Washüttl.

**Supervision:** Christian Kirchnawy.

**Validation:** Christian Kirchnawy.

**Writing – original draft:** Christian Kirchnawy, Fiona Hager, Johannes Mertl, Aurelie Mathieu-Huart, Christophe Rousselle.

**Writing – review & editing:** Fiona Hager, Johannes Mertl.

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
