## [Decision Letter · Decision Letter 0]

28 Jan 2020

PONE-D-19-32407

Potential endocrine disrupting properties of toys for babies and infants

PLOS ONE

Dear Mr. Rousselle,

Thank you for submitting your manuscript to PLOS ONE. After careful consideration, we feel that it has merit but does not fully meet PLOS ONE’s publication criteria as it currently stands. Therefore, we invite you to submit a revised version of the manuscript that addresses the points raised during the review process.

We would appreciate receiving your revised manuscript by Mar 13 2020 11:59PM. To enhance the reproducibility of your results, we recommend that if applicable you deposit your laboratory protocols in protocols.io, where a protocol can be assigned its own identifier (DOI) such that it can be cited independently in the future. For instructions see: http://journals.plos.org/plosone/s/submission-guidelines#loc-laboratory-protocols

We look forward to receiving your revised manuscript.

Kind regards,

Shanaz Hashmi Dairkee, Ph.D.

Academic Editor

PLOS ONE

Journal Requirements:

2. Please amend your manuscript to include your abstract after the title page.

3. Thank you for stating the following in your Competing Interests section:  "No"

Reviewers' comments:

Reviewer's Responses to Questions

**Comments to the Author**

1. Is the manuscript technically sound, and do the data support the conclusions?

Reviewer #1: Partly

Reviewer #2: Yes

2. Has the statistical analysis been performed appropriately and rigorously? 

Reviewer #1: N/A

Reviewer #2: N/A

3. Have the authors made all data underlying the findings in their manuscript fully available?

Reviewer #1: No

Reviewer #2: Yes

4. Is the manuscript presented in an intelligible fashion and written in standard English?

Reviewer #1: Yes

Reviewer #2: Yes

5. Review Comments to the Author

Reviewer #1: The topic of the manuscript is of high relevance. The data are worth to be published, although some questions remain open.

In the Chapter "Conclusions" is mentioned for the first time in the manuscript that BPA might not come from the investigated parts of toys, but also from the packaging (recycled cardboard boxes) and transportation. This opens the question why package material has not been investigated in this study to provide evidence for this speculation. At least citations related to this aspect should be implemented, i.e. Vandermarken et al. (2019): Chemosphere 221, 99-106 and references cited in there.

The desriptiom of the chemical analytical methods is insufficient and should be extended (see several remarks in the attrached file). It is refered to a previous publication (Mertl et al. (2014) in Plos-one). But in this publication many analytical details are also missing. Recovery rates for the quantified compounds have to be implemented in one table and it should be mentioned whether recoveries have been considered for calculation of the quantities.

I am astonished about the high values for detection limits and quantitation limits (in the µg/L ranges). The authors should explain why the limits are so bad in comparion to other studies using the same or similar analytical methods.

The data used for Figure 2 have to be presented in detail.

Some recent publications dealing with this topic have not been considered but should be considered.

Szczepańska N., Namieśnik J., Kudłak B. (2016): Assessment of toxic and endocrine potential of substances migrating from selected toys and baby products. Environmental Science and Pollution Research 23, 24890-24990.

Asimakopoulos A.G., Elangovan M., Kannan K., (2016): Migration of Parabens, Bisphenols, Benzophenone-Type UV Filters, Triclosan, and Triclocarban from Teethers and Its Implications for Infant Exposure. Environmental Science and Technology 50, 13539-13547.

Potouridis T. Knauz A., Berger E. & Püttmann W. (2019): Examination of paraben release from baby teethers through migration tests and GC–MS analysis using a stable isotope dilution assay. BCM Chemistry 13, 70. https://doi.org/10.1186/s13065-019-0587-6 and references cited in there.

Additonal comments are provided in the attached file.

Reviewer #2: Reviewer comments:

This study was designed to evaluate potential endocrine disrupting properties of toys for babies and infants by using in vitro cell-based reporter gene assays detecting ER and AR agonist and antagonist combined with chemical analysis. Authors clearly indicated that 9 out of 18 tested toys showed significant ER agonist activity. Among them, the detected ER agonist activity could be well explained by well-known EDC “BPA” for two samples. However, for seven samples, ER agonists, which strongly contribute to detected activity, could not be identified. Exposure assessment based on the worst-case scenarios suggest that BPA-containing toys could be above the temporary TDI.

Topic is very important and meaningful. However, there are several concerns in this manuscript. After revisions according to the following comments, manuscript will be accepted in PLOS ONE.

1. Abstract – I think authors had better describe about existence of unidentified ER agonists in toys in this section because they must be important on the toxicological point of view. I think that unidentified/unrevealed ER agonists are more important than BPA detected in this study. Although BPA will phase out, unidentified/unrevealed ER agonists might be emerging contaminants. Related to this point, authors should try to think and state importance of using in vitro bioassay detecting EDC. In vitro bioassay is not mere screening method and must be useful hazard detection method.

2. Materials and Methods, 1.1. Selection of toys, line 115 – Table 2 is Table 1?

3. Materials and Methods, 1.5. Chemical trace analysis, line 185 to 189 – Authors had better state why you measured these 41 known or suspected endocrine substances as well as common alternatives to endocrine active plasticizers in not only the text but also Table 1. Related laws or regulations are very welcome and must be useful for reader to understand importance of this study.

4. Materials and Methods, 1.7. Assessment of the health risks associated with the mouthing of plastic toys containing phthalate substitutes, line 243 to 246 – Authors should indicate “BW”, “F”, “S”, and “D” used in this study in the section of results and discussion by using Table or Supporting Information.

5. Results and discussion, Table 3 – Authors should indicate LOD with a figure for “Anti-estrogen activity” and “Anti-androgen activity”.

6. PLOS authors have the option to publish the peer review history of their article (what does this mean?). If published, this will include your full peer review and any attached files.

Reviewer #1: No

Reviewer #2: No

---

## [Author Response · Author response to Decision Letter 0]

16 Mar 2020

Responses to Reviewer #1: 

Comment 1 : « In the Chapter "Conclusions" is mentioned for the first time in the manuscript that BPA might not come from the investigated parts of toys, but also from the packaging (recycled cardboard boxes) and transportation. This opens the question why package material has not been investigated in this study to provide evidence for this speculation. At least citations related to this aspect should be implemented, i.e. Vandermarken et al. (2019): Chemosphere 221, 99-106 and references cited in there. »

Response : We agree with the reviewer. Therefore, we have included an additional paragraph (at the end of 3.3. Chemical trace analysis, line 441 - 457), explaining the card board packaging as a possible source of contamination, and also citing the suggested reference and two new references. In the conclusion section, storage and transport are now only mentioned briefly as possible source, as this has already been discussed before.

An additional analysis of the packaging would indeed have been interesting and could maybe have helped to better explain the source. However, this additional analysis would have been beyond the scope of the study, and it is also not possible anymore to be performed as packaging material is not available for testing anymore. For this study we wanted to investigate the toy as it is delivered to the customer which includes possible contaminations from packaging or transport.

Comment 2:

«The desriptiom of the chemical analytical methods is insufficient and should be extended (see several remarks in the attrached file). It is refered to a previous publication (Mertl et al. (2014) in Plos-one). But in this publication many analytical details are also missing. Recovery rates for the quantified compounds have to be implemented in one table and it should be mentioned whether recoveries have been considered for calculation of the quantities.»

Response : The description of chemical analysis was revised and explained in more detail (Chapter 2.5, starting at Line 231). Recovery rates, and variation coefficient of a 6-fold analysis of spiked saliva solvent and the correlation coefficient of the calibration curves are listed in table 3 for all substances of the target analysis. The data are presented analogous to the original PLOS one publication by Mertl et al., where data in ethanolic solutions was presented instead of the recoveries in saliva solvent. The recovery rates have not been considered for the calculation of the quantities as discussed in the answer on page 6.

Comment 3:

«I am astonished about the high values for detection limits and quantitation limits (in the µg/L ranges). The authors should explain why the limits are so bad in comparion to other studies using the same or similar analytical methods.»

Response : The limits of detection are certainly not the best detection limits that can be technically achieved, but based on our judgement they are sufficient for the purpose of this study. For the design of this study following requirements were defined for the sensitivity of the analytics :

- The methods had to be able to detect and quantify substances at the level of toxicological safety thresholds

- The methods had to be able to detect substances at concentration levels, which would give a positive result in the in-vitro tests, in order to allow a correlation between chemical and biological analysis. 

Both requirements can be fulfilled with the limits of detection achieved in this study. 

Previous studies, focused on chemical analytics alone did achieve better limits of detection, but in general they are in the same order of magnitude. Limits of detection are approximately in the same range, as e.g. in a recent study by Andaluri (2018), who reported a detection limit of 3 ng for bisphenol A in 10 ml extract of ispropanol wipes referring to 0.3 µg/L in the 10 ml extraction solution, which is very close to the 0.4 µg/l, that we achieved in our study. Asimakopoulos (2016) achieved a detection limit of 0.2 ng/ml for BPA and 0.17 ng/ml for methylparabene. This is better than the detection limits we achieved in the study: 0.4 ng/ml (or µg/l) for BPA and 0.7 ng/ml for Methylparabene, but still in the same order of magnitude. 

Many of the tested phthalates and other plasticizers are ubiquitously present in the environment and can be detected practically everywhere. With lower detection limits, it can be assumed that much more phthalates and other ubiquitously present substances could be detected. This would not necessarily help to estimate the contribution of toys to the total exposure, as these low, environmental contaminations would not be specific for toys.

Comment : « The data used for Figure 2 have to be presented in detail. »

Response : An additional table was introduced to the supplementary data, that lists the values that were used to create the figure. Further, the y-axis (E2 max) was explained in the figure legend.

Comment :

« Some recent publications dealing with this topic have not been considered but should be considered.

Szczepańska N., Namieśnik J., Kudłak B. (2016): Assessment of toxic and endocrine potential of substances migrating from selected toys and baby products. Environmental Science and Pollution Research 23, 24890-24990.

Asimakopoulos A.G., Elangovan M., Kannan K., (2016): Migration of Parabens, Bisphenols, Benzophenone-Type UV Filters, Triclosan, and Triclocarban from Teethers and Its Implications for Infant Exposure. Environmental Science and Technology 50, 13539-13547.

Potouridis T. Knauz A., Berger E. & Püttmann W. (2019): Examination of paraben release from baby teethers through migration tests and GC–MS analysis using a stable isotope dilution assay. BCM Chemistry 13, 70. https://doi.org/10.1186/s13065-019-0587-6 and references cited in there. »

Response to Reviewer: We absolutely agree with the reviewer. The three publications plus one additional new reference have been included in the introduction (lines 87 and 94)

Responses to Reviewer #2: 

Comment :

« 1. Abstract – I think authors had better describe about existence of unidentified ER agonists in toys in this section because they must be important on the toxicological point of view. I think that unidentified/unrevealed ER agonists are more important than BPA detected in this study. Although BPA will phase out, unidentified/unrevealed ER agonists might be emerging contaminants. Related to this point, authors should try to think and state importance of using in vitro bioassay detecting EDC. In vitro bioassay is not mere screening method and must be useful hazard detection method. »

Response: We absolutely agree, it is important to stress this directly in the abstract. Two additional sentences were added to the abstract to describe about the existence of unidentified endocrine active substances, as was suggested.

Comment :

« 2. Materials and Methods, 1.1. Selection of toys, line 115 – Table 2 is Table 1? »

Response: The order of the tables was indeed unnecessarily confusing. This Table is now Table 1. The original Table 1 was extended by validation results, moved to the Results section and is now Table 3.

Comment :

« 3. Materials and Methods, 1.5. Chemical trace analysis, line 185 to 189 – Authors had better state why you measured these 41 known or suspected endocrine substances as well as common alternatives to endocrine active plasticizers in not only the text but also Table 1. Related laws or regulations are very welcome and must be useful for reader to understand importance of this study. »

Response: The selection of the 41 substances was better explained at the beginning of Chapter « 2.5. Chemical trace analysis ». It was difficult to include this information in the Table, because this table was already extended by validation parameters based on the feedback from the other reviewer, and the table would have become confusing and overloaded. We looked on several drafts for this table, that include this information, but decided at the end to put this information in the text rather than in this table.

Comment :

4. Materials and Methods, 1.7. Assessment of the health risks associated with the mouthing of plastic toys containing phthalate substitutes, line 243 to 246 – Authors should indicate “BW”, “F”, “S”, and “D” used in this study in the section of results and discussion by using Table or Supporting Information.

Response: We agree that this information was indeed missing in our draft. As the same values were used for all calculations, we did include this information directly in the explanation of the calculation. (Lines 288 – 300) 

Comment :

« 5. Results and discussion, Table 3 – Authors should indicate LOD with a figure for “Anti-estrogen activity” and “Anti-androgen activity”. »

Response: We absolutely agree, that LODs for the antagonists are missing here. We have added LODs for anti-estrogen activity (expressed as 4-OHT equivalents) and for anti-androgen activity (expressed as flutamide equivalents). A few additional sentences on the quantification of antagonistic activity by comparing to activities of the reference standards 4-OHT and flutamide was introduced in Chapter 2.4. While revising this part, it was noticed that the quantification of agonistic activity as estrogen equivalents (EEQ) and androgen equivalents (AEQ) was not well described in the original manuscript. Therefore, a better description was added here as well. 

Responses to Comments in the Manuscript

Comment Line 262: „Have the revalidation experiments been carried out on the same sample set that has already been used by Mertl et al. im 2014?

At which time were the samples obtained from the suppliers?”

Response: No, the revalidation experiment for the solvent concentration by solid phase extraction was performed using just saliva solvent, without further evaluating the influence of sample matrix by dissolved substances from the toys. In the previous study by Mertl et al. much higher amounts of dissolved oligomers and low molecular weight polymer had to be expected due to the much harsher extraction conditions (e.g. 10 d, 60°C, 95% ethanol compared to 30 minutes with an aqueous buffer at room temperature). Therefore, in the study by Mertl et al. (2014) the recoveries were determined in the presence of a representative worst-case sample matrix to evaluate if this high amount of dissolved polymer could negatively influence the recovery of target substances (e.g. by overloading the SPE column). As it has been demonstrated in this validation, that even at these harsh conditions and for a worst-case sample matrix the method was working in the presence of the sample matrix, it was considered that no further investigations are necessary to evaluate the influence of dissolved polymer again for saliva solvent. Instead the focus of the revalidation was, to evaluate if the method, that was previously applied for ethanolic solutions can also be used for the concentration of saliva solvent. The description of the revalidation was revised and explained in a bit more detail (see line 315 - 319).

Comment Line 185 “In Mertl et al (15) no details regarding GC/MS-analysis of the compounds were presented: GC-temperature program, heating rate...”

Response: The requested details have been included.

Comment Line 205 “This desription of methods is insufficient and no reference is provided indicating where more details were reported previously.”

Response: The description of the method was revised and is presented in more detail now. 

Comment Line 214 “you should mention the recoverys for those compounds that were quantified.”

Response: The recoveries are now listed in the revised table.

Comment Line 408. “Have quantities been corrected by recoveries?”

Response: No, recoveries were just determined to evaluate the reproducibility of the method but were not used to correct quantities. Recoveries ranged from 87% to 104%, which was not considered as necessary to correct quantities by recoveries. 

Comment Line 464: “The compounds must not be volatile to be transfered to saliva simulants.”

Response: This sentence referred to the fact that only volatile or semi-volatile compounds can be detected by GC-MS, while completely involatile compounds might be responsible for the bioassay activity. It was clarified, and changed to “is not detectable by the applied GC-MS method “(see line 545 in the revised manuscript)

The other comments in the manuscript were directly addressed and do not require any further argumentation according to the judgement of the authors.

---

## [Editor Report · Decision Letter 1]

18 Mar 2020

Potential endocrine disrupting properties of toys for babies and infants

PONE-D-19-32407R1

Dear Dr. Rousselle,

We are pleased to inform you that your manuscript has been judged scientifically suitable for publication and will be formally accepted for publication once it complies with all outstanding technical requirements.

With kind regards,

Shanaz Hashmi Dairkee, Ph.D.

Academic Editor

PLOS ONE
---

## [Editor Report · Acceptance letter]

23 Mar 2020

PONE-D-19-32407R1 

Potential endocrine disrupting properties of toys for babies and infants 

Dear Dr. Rousselle:

I am pleased to inform you that your manuscript has been deemed suitable for publication in PLOS ONE. Congratulations! Your manuscript is now with our production department. 

With kind regards,

on behalf of

Dr. Shanaz Hashmi Dairkee 

Academic Editor

PLOS ONE